# Knot So Simple:
# A Minimalistic Environment for Spatial Reasoning

**Zizhao Chen and Yoav Artzi**
Department of Computer Science and Cornell Tech, Cornell University
{czz, yoav}@cs.cornell.edu

## Abstract

We propose KNOTGYM, an interactive environment for complex, spatial reasoning and manipulation. KNOTGYM includes goal-oriented rope manipulation tasks with varying levels of complexity, all requiring acting from pure image observations. Tasks are defined along a clear and quantifiable axis of complexity based on the number of knot crossings, creating a natural generalization test. KNOTGYM has a simple observation space, allowing for scalable development, yet it highlights core challenges in integrating acute perception, spatial reasoning, and grounded manipulation. We evaluate methods of different classes, including model-based RL, model-predictive control, and chain-of-thought reasoning, and illustrate the challenges KNOTGYM presents. KNOTGYM is available at `https://github.com/lil-lab/knotgym`.

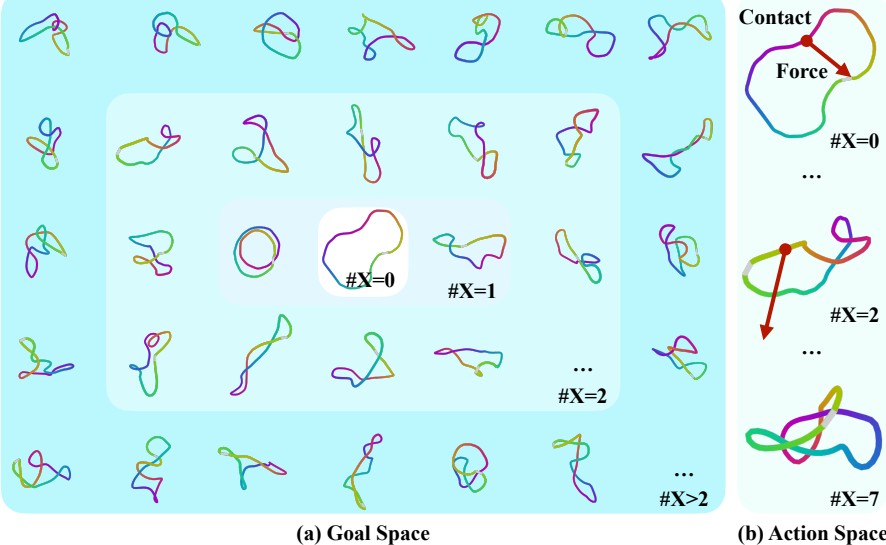

(a) Goal Space           (b) Action Space

Figure 1: KNOTGYM is a visual reasoning knot manipulation environment. It includes three tasks: transforming complex knots to a simple loop (example in the center); tying a loop into complex knots (right pane); and converting one knot into another, given a goal knot image. KNOTGYM has a continuous visual observation space (left pane) and an action space of applying forces to contact points (right pane), abstracting the specifics of robot end effectors. The space of goals, specified using Gauss code, is a factorial of the number of crossings (#X), which creates a ladder of generalization. Each goal defines an easily testable equivalence class over a continuous set of states.

# 1 Introduction

KNOTGYM is a knot manipulation environment to train and evaluate visual reasoning agents. Figure 1 illustrates KNOTGYM. The agent observes the world state as an image and needs to transform a knot into a topological goal by applying forces to rope segments. The goal is communicated to the agent by observing a secondary knot as an exemplar. KNOTGYM defines three tasks (Table 1), each characterized by its start state and goal. KNOTGYM is designed with several aspects in mind:

**Research Challenges** KNOTGYM evaluates spatial reasoning, action prediction, planning, and abstraction skills in a single environment. Solving a task requires analyzing the current rope configuration, planning on deforming it to achieve the goal, and mapping this plan to a sequence of continuous action predictions. Critically, KNOTGYM does not enforce one specific state as the goal – the goal is a large equivalence class of states. Goals are defined via Gauss code, a formal mathematical description of knot configuration based on its crossings. When the goal is communicated using an exemplar image, solving a task requires abstracting over the exemplar to identify the right set of actions to complete the task. We show that KNOTGYM is a significant challenge for contemporary methods.

**Measurable Complexity** Knots are well studied in mathematics, with formal descriptions like Gauss code and a measure of complexity in the number of crossings.[1] Figure 1 shows various knot configurations with different numbers of crossings, illustrating the increasing complexity of more crossings. This measure of complexity enables to control the challenges of learning and evaluation.

**Generalization Ladder** The number of crossings creates a clear generalization ladder. This allows a clear train-test complexity split, where we train up to a certain level of complexity (i.e., number of crossings) and test if a model generalizes beyond this number. During training, this allows us to experiment with a natural learning curriculum, where the number of crossings in training examples is gradually increased. It also enables the study of self-improving bootstrapping processes in a visual domain, as recently proposed for arithmetic and maze-solving [Lee et al., 2025b].

**Research Accessibility** KNOTGYM is an accessible task particularly suited for studying extended visual reasoning in a laboratory environment, just as Pendulum for classic control and Countdown for reasoning in natural language [Gandhi et al., 2024]. We follow the implementation and design principles of OpenAI Gym [Brockman et al., 2016] to make the KNOTGYM as accessible for researchers as possible. This includes following the standard Gymnasium API [Towers et al., 2024] and supporting vectorized environments on multiple CPUs. The KNOTGYM environment and our baselines are available under the MIT license at `https://github.com/lil-lab/knotgym`.

Table 1: The three tasks of KNOTGYM. We show the final observation assuming the goal is achieved. Correctly completing tasks does not require getting the exact exemplar configuration (right rope in each image pair), but a configuration with the same Gauss code, a notation that describes the abstract spatial characteristics of a knot. For example, the final observations in the `tie` row are both [1+,1-,2+,2-], despite their different appearances.

| Task name | Description | Initial→ Final Observations |
|---|---|---|
| `unknot` | Untangle a knot into a simple loop |  |
| `tie` | Tie a goal knot from a simple loop |  |
| `convert` | Tie a new knot from an old knot |  |

---

[1]While the number of crossings is a fitting measure of complexity to our manipulation task, the prime complexity measure of interest in mathematics is more likely the counting of crossing of *irreducible knots* (i.e., as part of the tabulation of prime knots). Under this perspective, all the knots in Figure 1 are equivalent, because they all can be reduced to a simple loop by applying a few (or many) Reidemeister moves. For us, the number of crossings is a more appropriate measure because we aim for manipulation and observation complexity.

## 2  Related Work

**Spatial Reasoning in Vision-language Models**  Spatial reasoning has been studied with benchmarks focused on language-vision tasks, specifically relations between individual objects using synthetic [Johnson et al., 2017, Suhr et al., 2017] and natural [Suhr et al., 2019, Liu et al., 2023] images. The evaluation of the spatial reasoning of agents is often embedded in locomotion or robotics manipulation tasks [Yang et al., 2025, Shridhar et al., 2020]. These tasks prioritize diverse domains and spatial relationships, but they are static and require limited reasoning. In contrast, KNOTGYM, is more narrow, but demands long, complex spatial reasoning, opening up opportunities to apply test-time-scaling reasoning in a visual domain. Similar narrow focus to KNOTGYM is deployed by lilGym [Wu et al., 2023], but with focus relations between rigid objects expressed in natural language.

**Axes of Generalization**  Generalization is studied along many axes, such as length in text sequence models [Anil et al., 2022], size in graph neural networks [Yehudai et al., 2021], combining parts in new ways [Hupkes et al., 2020], performing arithmetic [Lee et al., 2025b, Dziri et al., 2023], and symbolic operations [Welleck et al., 2022]. KNOTGYM proposes a new *visual* generalization axis. We leverages the well-studied mathematical construct of knots to provide an established formulation to structure and discuss generalization evaluation.

**Deformable Object Manipulation**  Manipulating deformable objects, such as liquid, playdough, and ropes, is notorious for its infinite dimensional configuration space [Lin et al., 2020]. Existing work [Yan et al., 2020, Sundaresan et al., 2021, Shi et al., 2024] focuses on learning task-specific representations, and aims to generalize robustly across different textual and material in the real world. KNOTGYM takes the idea of rope manipulation and simplifies the apparatus to emphasize complexity generalization. This allows us to assess the complex visual reasoning abilities of both RL methods and general pretrained VLMs. KNOTGYM also has a different goal formulation compared to classic robotics tasks. Our goal is to reach an abstract topological structure determined by the Gauss code, as opposed to minimizing a distance (i.e., Euclidean) between one set of coordinates and the goal coordinates. The latter can be formulated as an optimization problem, while the former relies more on abstract reasoning (even a form of searching).

**Machine Learning (ML) and Knots**  The intersection of ML and knot theory research is limited, but promising. Davies et al. [2021] discovered a new connection between the algebraic and geometric structure of knots, using ML to guide human intuition. Part of KNOTGYM is an instantiation of a knot-theoretic problem called unknotting, a special instance of the generic and open problem of knot equivalence. To solve the same unknotting task, Gukov et al. [2021] learns a reinforcement learning powered search algorithm in symbolic space via braid words. In contrast, we focus on raw image observations and are interested in assessing a model's ability to reason about intuitive physics.

## 3  KnotGym

Intuitively, knots are ropes whose ends are joined. In KNOTGYM, agents manipulate such objects in a continuous 3D space by pulling on specific points in the rope (i.e., exerting force at a location), without breaking the continuity of the rope. Each knot is embedded in 3D space. The 3D coordinates of a knot are its *configuration*. Each episode has a topological goal expressed by an underlying Gauss code. The goal is specified via an exemplar placed in the environment, adjacent to the manipulated knot. The goal of each episode is to manipulate an initial knot configuration, via a series of actions, such that the final knot has the goal Gauss code (Figure 2). The agent in KNOTGYM does not have access to the world state or the Gauss code representing the goal, but instead receives visual observations only, reflecting our research interest in visual spatial reasoning. KNOTGYM can be easily extended to reveal those signals to the agent as well.

We now define and discuss the design of the KNOTGYM environment and tasks. Table 2 summarizes key terms. KNOTGYM implements the Gymnasium interface [Towers et al., 2024] for ease of use. Appendix C provides implementation details.

Table 2: Key concepts in KNOTGYM

| Concept | Explanation |
| --- | --- |
| (Knot) configuration | 3D coordinates of key points along a single knot |
| Goal | Abstract spatial relationship of a knot uniquely identified by a Gauss code (many configurations can share the same Gauss code) |
| Goal configuration | A knot configuration that has the goal Gauss code |
| State | The environment includes both the manipulated knot and the goal, so the state specifies both of their configurations |
| Observation | An RGB image that is a 2D projection of the state onto the z-plane |

## 3.1 Environment

KNOTGYM is an episodic partially observable Markov decision process (POMDP) $(\mathcal{S}, \mathcal{A}, \mathcal{T}, \mathcal{R}, \Omega, \mathcal{O}, H, \gamma)$, where $\mathcal{S}$ is the state space, $\mathcal{A}$ is the set of actions, $\mathcal{T}$ is the transition function, $\mathcal{R}$ is the reward function, $\Omega$ is the observation space, $\mathcal{O}$ is a mapping between states and image observations,[2] $H$ is a termination criteria, and $\gamma$ is the discount factor.

**States** $\mathcal{S}$   A state $s_t \in \mathcal{S}$ in time $t$ is a pair $(c_t^m, c^g)$ of two rope configurations, $c_t^m$ is the current configuration of the manipulated rope, and $c^g$ is the configuration of the goal exemplar. Knot configurations are represented by a series of 3D coordinates of key points along the rope. The goal configuration $c^g$ satisfies the goal Gauss code and remains the same throughout an episode; we include it in the state space to construct a Markovian state and to keep the policy conditioning on the state only. The agent operating in KNOTGYM does not have access to the world state, but receives partial observations.

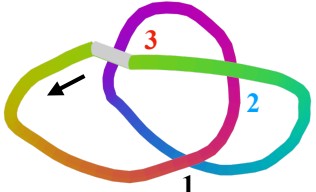

**Gauss code: 1+,2-,3-,1-,2+,3+**

Figure 2: An episode is successful when the current knot configuration has the goal Gauss code. We obtain the Gauss code of any knot by traversing through the rope, starting from the white segment towards red (black arrow). When traversing, we denote an over-cross with +, and an under-cross with -, until we return to the starting segment.

**Observations** $\Omega$   An observation $o_t \in \Omega$ is an image generated by the observation mapping $\mathcal{O} : \mathcal{S} \to \Omega$. Concretely, $o_t$ is rendered z-plane projections of both the current configuration $c_t^m$ and the goal configuration $c^g$. Observation images have the shape $[3, 128, 2 \times 128]$ because we concatenate the rendered images of both configurations. Similar to [Lin et al., 2020], we use a larger image size than canonical deep RL environments because crossings are critical to determine the Gauss code, which is critical for the agent's reasoning. Crossings and self-occlusions also contribute to the partial observability of this problem. We reduce self-occlusions by tuning cable diameter and increasing the resolution.[3] We use 2D RGB observations because we are inspired by how humans perform predictive spatial reasoning from visual inputs only. KNOTGYM can be easily extended to symbolic observations or multiple cameras or RGBD observations.

**Actions** $\mathcal{A}$   An action $a \in \mathcal{A}$ is a 6-tuple $(x, y, z, f_x, f_y, f_z)$, where $(x, y, z)$ specifies a 3D location in state space that is rounded to the closest key point on the manipulated rope, and $(f_x, f_y, f_z)$ is a force vector to apply to that key point. The action vector is tuned and normalized to $[-1, 1]^6$. Our main focus is spatial reasoning, so we abstract away end effectors that would introduce hardware-specific policies.[4] The transition model $\mathcal{T} : \mathcal{S} \times \mathcal{A} \to \mathcal{S}$ captures the change in knot coordinates after applying the force for a short period of time, which is implemented by the MuJoCo physics simulator.

---

[2]KNOTGYM has a largely deterministic mapping from states to observations, even though there is negligible stochasticity because of the simulator.

[3]Cable diameter and resolution should be tuned when scaling to more complex knots (#X>8).

[4]Training with realistic end effectors is easy to add in the MoJoCo ecosystem, although tangential to our focus on abstract spatial reasoning.

**Reward Function** $\mathcal{R}$    The reward function $\mathcal{R} : \mathcal{S} \to \mathbb{R}$ is sparse and based on a symbolic oracle $GC$, which maps the rope coordinates to its Gauss code. The reward function is defined as

$$\mathcal{R}(c^m, c^g) = \mathbb{1}(GC(c^m) = GC(c^g)) \ .$$

The agent receives a single positive reward if the current knot configuration has the same Gauss code as the goal configuration. The agent needs to assess the Gauss code itself. The Gauss code is a formal description of the topology of a knot. It is computed by traversing through the rope and recording under-/over-crossings (Figure 2). Gauss codes are necessary but not sufficient to fully reconstruct a knot.[5] Table 3 shows how #X determines the number of possible Gauss codes. In addition, the reward function provides a negative penalty when reaching the horizon limit without success.

This criterion for success introduces a level of abstraction into KNOTGYM, where an agent must abstract the full set of correct termination states from the exemplar goal image. It is also more lenient than requiring reconstruction of the exact goal configuration, and as our experiments show, already challenging enough for existing methods (Section 4). Because the reward is based on topological equivalence, it does not always correspond to visual distance-based measures: two rope configurations can be completely different at first glance yet share the exact same underlying Gauss code; conversely two visually similar configurations can differ in their Gauss codes by just changing a single over-cross to an under-cross. This choice distinguishes KNOTGYM from other rope-configuration tasks, such as SoftGym [Lin et al., 2020], where dense rewards are computed from coordinate-wise distance that easily enable trajectory optimizations. KNOTGYM, in contrast, requires global and holistic spatial reasoning. It is not immediately clear, at least not to us, how to convert the tasks into smooth optimization problems.

**Termination Criteria** $H$    There are two criteria for termination: completing the task (i.e., equal Gauss code, which also entails a positive reward) or reaching the horizon limit. An episode ends immediately upon reaching the goal Gauss code. An alternative, stricter reward function is to maintain the goal Gauss code for multiple frames (i.e., over a set time period), effectively learning a stopping behavior. Our preliminary experiments suggest that this variation makes the tasks even more difficult. It is easy to add to KNOTGYM.

Table 3: The factorial space of Gauss codes (GCs) with respect to the number of crossings (#X).

| #X | # Possible GCs | Example GCs |
|---|---|---|
| 0 | 1 | { [] } |
| 1 | 2 | { [1+, 1-], [1-, 1+] } |
| 2 | 12 | { [1+, 1-, 2+, 2-], [1+, 1-, 2-, 2+], [1-, 1+, 2+, 2-] ... } |
| $n$ | $(2n-1)!!2^n$ | ... |

### 3.2 Three Tasks

We design three tasks: `unknot`, `tie`, `convert`. They follow exactly the same environment interface as defined above, while only differing in the selection of initial state and the goal Gauss code, see Table 1. The complexity of each task can be tuned by setting the number of crossings in the initial rope configuration or the goal. For example, Table 4 shows the crossing settings we use in our experiments. We collect 40 knot configurations for each crossing setting (17 for the simple loop #X=0), reserve 20 configurations as

Table 4: Three tasks based on the number of crossings (#X) of the initial/goal knots.

| Task name | Initial #X | Goal #X |
|---|---|---|
| unknot | {2,3,4} | {0} |
| tie | {0} | {2,3,4} |
| convert | {1,2,3} | {2,3,4} |

the train split, and sample the initial goal configurations randomly at training and testing time. We set the horizon limit to 50 steps in our experiments to balance exploration and training costs. This is a hyperparameter that users can modify if they increase the number of crossings.

The easiest of the three tasks is `unknot`, which requires untangling a knot into a simple loop (goal #X=0). The goal of `unknot` is shared across all episodes, so the policy does not have to reason about differing goals. `tie` is the inverse of `unknot`: it requires tying a knot from a simple loop. The policy is goal-conditioned: it needs to uncover the underlying goal Gauss code from an image, and

---

[5]An extended variation of Gauss code records additional chiral information.

manipulate the simple loop towards the identified goal Gauss code. `convert` is a combination of the other two and requires a wide range of abilities: identifying the goal Gauss code, recognizing the current topological structure, and planning accordingly.

### 3.3 Evaluation and Generalization

The primary evaluation metric is success rate: the percentage of episodes that with a positive reward (i.e., the manipulated knot configuration matches the goal Gauss code).

KNOTGYM offers at least two testable axes of generalization. The first is train/test generalization: we train the policy on initial/goal configurations from one split, and evaluate on unseen test configurations, keeping the task and the number of crossings #X the same. The second, and harder axis is complexity generalization, when the policy is trained on configurations up to a certain number of crossings (e.g., #X=2) and tested on configurations with a higher number (e.g., #X=3 and beyond). This notion of generalization is related to length generalization [Lee et al., 2025b, Dziri et al., 2023] or size generalization Yehudai et al. [2021]. A policy that generalizes well will succeed at the task even when the knot configurations are completely new and more complex. For training-free baselines (i.e., prompting proprietary VLMs), we assume that KNOTGYM is out of the pretraining and post-training distribution, and therefore it is always generalizing its training distribution.

### 3.4 Knot So Simple

What makes KNOTGYM both interesting and challenging? We identify three unique features KNOT-GYM offers. These features are tightly integrated, making it suitable to test end-to-end systems.

**Acute Perception**    Unlike the tasks in SoftGym [Lin et al., 2020], solving KNOTGYM relies heavily on correct identification of crossings, which often span only a few pixels. Effective policies must attend to such minute details to reason about both the goal and the current topological state.

**Continuous Spatial Reasoning**    Unlike environments such as Ant-Maze [Fu et al., 2020], or ARC-AGI [Chollet, 2019], which have a discrete reasoning space, KNOTGYM, by its definition (a closed loop embedded in $\mathbb{R}^3$), is a naturally continuous environment with continuous states and action sets. There is no clear unit of action that would mark a branching point to enable the application of discrete planning methods. While it is conceptually possible to discretize the knot coordinates, for example, as a link diagram, and search over Reidemeister moves to achieve the goal Gauss code, KNOTGYM would still require mapping to grounded actions back in the continuous space.

**Very Large Search Space**    The search space is large not only because KNOTGYM is a continuous space and the set of possible Gauss codes grows factorial with the number of crossings, but also because the knot equivalence problem is known to be hard. For the scope of this project, all knot configurations are reducible to the trivial knot, or the so-called unknot. Essentially, the policies are required to prove this by concretely finding a reduction path. As our experiments show, this already proves a significant challenge to state-of-the-art methods (Section 4). Whether we can decide if an arbitrary knot is reducible to a simple loop in polynomial time is an open math problem [Burton and Ozlen, 2012]. What's more, KNOTGYM can easily extend to include non-trivial knots, such as the trefoil and its variants. The north star of KNOTGYM is to drive the development of policies that learn to solve the knot equivalence problem, a generalization of the unknotting problem.

## 4 Experiments

We evaluate representative methods of different types on KNOTGYM: PPO (model-free RL) [Schulman et al., 2017], DreamerV3 (model-based RL) [Hafner et al., 2025a], TM-MPC2 (RL with test-time search) [Hansen et al., 2024], and VLMs via chain-of-thought prompting using GPT-4.1-nano [Achiam et al., 2023]. Our results characterize the strengths and weaknesses of each method. We discuss RL and prompting methods separately, as the experiments bring different insights.

Table 5: Benchmarking representative methods over nine KNOTGYM setups. Entries are training split success rates calculated over $N$ rollouts. For RL, measurements are taken at 1M steps.

| Task | #X | Random | RL ($N$=384) | | | Prompting ($N$=105±7) | | |
|---|---|---|---|---|---|---|---|---|
| | | ($N$=256) | DreamerV3 | PPO | TD-MPC2 | Open | Stateless | Stateful |
| `unknot` | 2 | 11.1 | 93.3 | 65.7 | 71.3 | 0.0 | 12.0 | 20.9 |
| `unknot` | 3 | 8.4 | 93.4 | 63.3 | 55.4 | 0.0 | 6.7 | 17.0 |
| `unknot` | 4 | 6.7 | 89.3 | 37.0 | 50.3 | 0.0 | 6.1 | 7.4 |
| `tie` | 2 | 35.9 | 83.2 | 41.3 | 39.2 | 0.0 | 7.6 | 5.5 |
| `tie` | 3 | 2.5 | 16.1 | 3.3 | 4.6 | 0.0 | 1.0 | 0.9 |
| `tie` | 4 | 1.4 | 4.1 | 1.3 | 1.7 | 0.0 | 0.0 | 0.0 |
| `convert` | 2 | 36.8 | 71.5 | 47.7 | 40.8 | 0.9 | 5.7 | 2.8 |
| `convert` | 3 | 9.2 | 15.3 | 6.3 | 8.7 | 0.0 | 1.0 | 0.9 |
| `convert` | 4 | 2.9 | 5.4 | 4.7 | 3.8 | 0.0 | 0.0 | 0.0 |

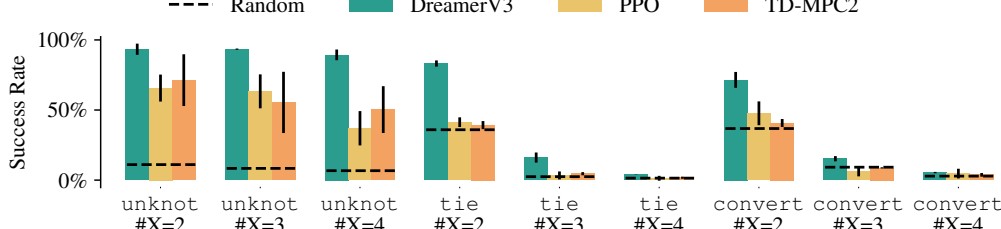

Figure 3: Train success rates of RL methods on nine different KNOTGYM setups after 1M environment steps during training. Error bars represent 95% confidence interval. All methods show non-trivial improvements on `unknot` via RL training, but struggle on `tie` and `convert`. No methods outperform a random policy at #X=4 of `tie` and `convert`, suggesting that increasing #X raises task difficulty significantly for tasks with many possible goals.

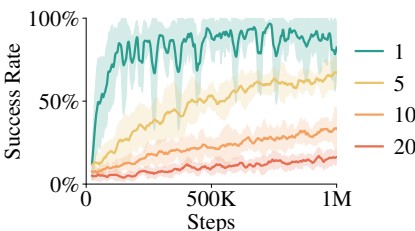

Figure 4: Training curves for different number of goal configurations in the training set (DreamerV3, `tie`, #X=3).

Figure 5: Generalization matrices for three tasks. Each entry of the matrices is success rate evaluated on the test split with $N$=128 episodes.

## 4.1 Reinforcement Learning Methods

We experiment with the vision variant of PPO implementation in Stable-Baselines3 [Raffin et al., 2021], and the official implementations of TD-MPC2 and DreamerV3, following the default configurations as much as possible. We include detailed hyperparameters in Appendix B and open-source benchmarking code for reproducibility. Each RL training run has a fixed budget of 1M environment steps. Table 5 and Figures 3–6 present the results and analysis. Appendix D presents additional results including all training curves in Figure 13.

**Comparing the Three Tasks** All methods show non-trivial performance on the easiest `unknot` task after training, with DreamerV3 performing particularly well. However, results on `tie` or `convert` are significantly weaker. A potential explanation is that `unknot` is slightly simpler because the goal is always the same, and the agent does not need to decode the Gauss code from the goal exemplar. Qualitative analysis (Figure 6) shows that all methods recover a simple strategy to solve `unknot`: dragging one section of the rope in one direction and letting inertia untangle the knot. This generalizes well to more crossings. In contrast, `tie` and `convert` require much more careful reasoning about the

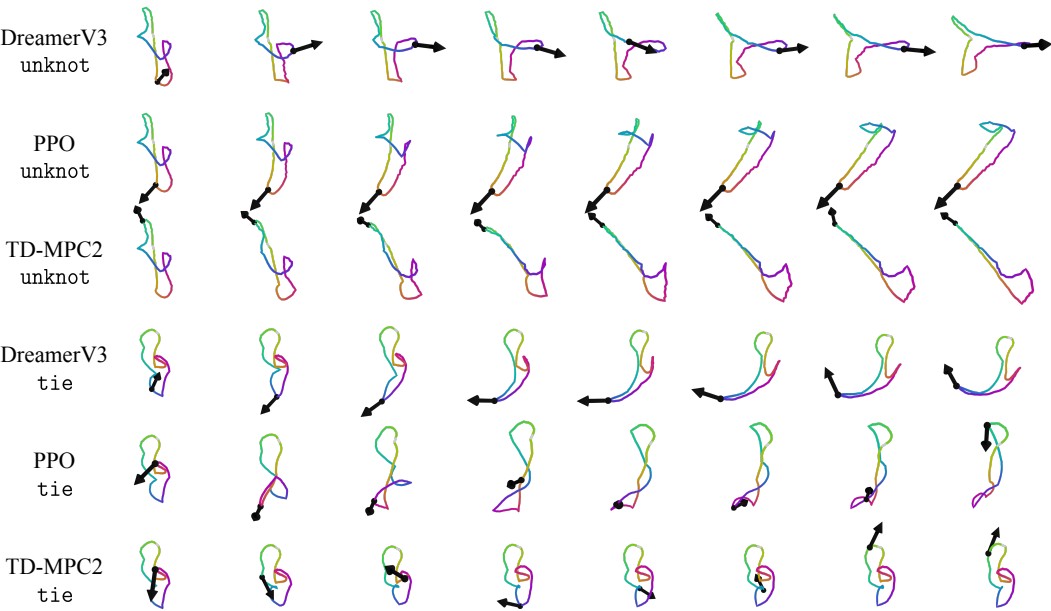

Figure 6: RL policy rollouts on `unknot` and `tie`. All methods find a solution to `unknot` by consistently pulling (black arrows) one segment of the rope. However, for `tie` the actions are more diverse and spread out without a consistent pattern, highlighting the difficulty of `tie`.

goal and condition the policy on the decoded goal Gauss code per episode, which contributes to the difficulty of `tie` and `convert`.

**Training Difficulty and the Number of Crossings (#X)**    Increasing the number of crossings raises task difficulty significantly for `tie` and `convert`. Our random baseline policy samples actions uniformly. The random baselines for #X=2 solve a significant number of tasks because random walking for 50 timesteps likely stays in the space of #X=2. This is likely helpful for exploration early on during RL training for `convert` at #X=2. At #X=3, only DreamerV3 learned a policy clearly better than random for `tie` and `convert`. With one more crossing (#X=4), all methods are merely marginally better than random policy at `tie` and `convert`, even on the training tasks. This highlights the training challenges with larger #X. We hypothesize this challenge is related to the size of the training pool, so we examine the training curves when we constrain the diversity of goal configurations seen during training, reducing from the default 20 goal configurations to 10, 5, and 1 (Figure 4). Indeed, the variety of goal configurations presents a challenge for RL training.

**Generalization**    We train three DreamerV3 policies for `unknot` and `tie` on a pool of 20 initial configurations with #X=2,3,4 respectively, and evaluate on held-out test initial configurations with #X=2,3,4. Figure 5 shows generalization performance. `unknot` policies generalize well: policies trained on #X=2 work well on new #X=2 knots, and generalize to new #X=3 knots, and even, to a lesser degree, to new #X=4 knots. For the harder `tie`, the #X=2 policy, despite reaching over 80% training success rate, only narrowly outperforms the random policy on the test set of new #X=2 knots, suggesting overfitting to training examples. There is also little goal-generalization on `tie`, as the test success rates are close to, if not lower than, the respective random policies.

## 4.2   Prompting

We evaluate GPT-4.1-nano on KNOTGYM. Because it is fair to assume that KNOTGYM is a new task for current VLMs, prompting naturally evaluates train/test and complexity generalization.[6] Each prompt is prefixed with detailed instructions about the task, definition of Gauss code, and tuples of before-after comparisons to convey system dynamics. We experiment with several prompting modes per task, with 100 episodes per task-mode pair.

---

[6]Early experiments show that "thinking" models (`o1`) do not have an edge over regular models in KNOTGYM.

Figure 7: Example prompt and response in an open mode query. The VLM recognizes the goal Gauss code and forms a reasonable plan to reach the goal. However, during rollouts the actions are too weak to achieve the desired effect, highlighting the brittleness of naive world modeling in VLMs.

**Prompting Modes**  We evaluate three prompting modes. They differ in the control loop (open versus closed) and whether memory is retained for future queries: (a) *open* mode: the VLM produces a sequence of actions in one shot, given the initial observation $o_0$, including images of both current $c_0^m$ and goal $c^g$ configurations. (b) *stateless* mode: the VLM produces one action at a time, given the current observation $o_t \sim \mathcal{O}(c_t, c^g)$; and (c) *stateful* mode: the VLM also produces one action at a time, but it observes a windowed history of past observations and actions. The open/closed control loop (open vs. stateless/stateful) offers an opportunity to assess VLM dynamics without seeing the effects of its own actions. The stateful/stateless comparison is based on the assumption that memory would help further contextualize the system dynamics. Prompt examples are included in Appendix A.

**Results**  Prompting-based methods perform worse than RL baselines in general (Table 5). Most prompting baselines are worse than a random policy, except for unknot, #X=2,3. The observation history offered by stateful prompting seems to help, as evidenced on the unknot task. Figure 7 presents one example VLM response to the open query. The VLM recognizes the goal Gauss code is [] (the task is unknot) and forms a reasonable plan to untangle the knot. However, during rollouts, the actions are too "weak" and not precise enough to achieve the desired effect. Action imprecision issues persist even with closed-loop feedback, highlighting the inefficiency of naive world modeling using VLMs to perform fine manipulations.

## 5  Discussion

We present KNOTGYM, a new interactive environment for complex spatial reasoning and manipulation. We benchmark general-purpose RL methods of different classes, and show a gradient of difficulty: unknot is relatively easy, but tie and convert pose a significant generalization challenge to state-of-the-art RL methods. Key to enabling this observation is the well-defined generalization ladder knot theory provides. KNOTGYM also presents a third type of generalization underexplored in this work that can be informally called *causal generalization*. For example, we can train the policy on tie task (the forward direction) trajectories, and evaluate whether the policy generalizes to the unknot task in the backward direction.

KNOTGYM admits a broad range of solutions. RL methods posed by the vast goal space generally struggle with data inefficiency. Prompting VLMs with curated prompts, while succeeding at understanding the goal and performing some spatial reasoning, cannot produce fine, grounded actions. Including multi-turn interaction history in the prompt helps only to a limited extent. KNOTGYM provides an excellent testbed for evaluating other frontier visual reasoning agents, for instance, agents that verbalize action space reasoning [Lee et al., 2025a, MolmoAct], agents that rollout within the learned latent model [Hafner et al., 2025b, Dreamer4] or in image space based on unified multimodal models [Wu et al., 2024, VILA-U], agents that leverages pretrained video models as world models [Ball et al., 2025, Genie3], even visual coding agents that construct an explicit physics-based model of the rope through interaction.

KNOTGYM is now ready to enable the research of a broad set of research questions. We continue to develop it, with a plan to address several limitations in the near future. The first is simulator capacity, including improving simulation fidelity (e.g., to improve scaling to a high number of crossings and longer ropes), and throughput (e.g., currently 20 environment steps per second on a single CPU after applying frame skip) by moving from CPUs to GPUs.

## Acknowledgments and Disclosure of Funding

This research was supported by NSF under grants No. 1750499, a gift from Open Philanthropy, an Nvidia Academic Grant, the National Artificial Intelligence Research Resource (NAIRR) Pilot, the Frontera supercomputer supported by the National Science Foundation (award NSF-OAC 1818253) at the Texas Advanced Computing Center (TACC) at The University of Texas at Austin, and the Delta advanced computing and data resource which is supported by the National Science Foundation (award NSF-OAC 2005572). We gratefully acknowledge use of the research computing resources of the Empire AI Consortium, Inc, with support from the State of New York, the Simons Foundation, and the Secunda Family Foundation [Bloom et al., 2025]. We thank Haochen Shi for helpful discussions and proofreading, and Steve Marschner and Kuan Fang for advice on 3D simulation.

Any opinions, findings and conclusions or recommendations expressed in this material are those of the author(s) and do not necessarily reflect the views of the National Science Foundation, NASA, or the other funders.

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

# A Prompt Examples

We provide the prompt instances for each prompting mode: open prompt (Figure 8), stateless prompt (Figure 9), and stateful prompt (Figure 10). The first image of the user prompt (i.e., first <image> tag) is the same across the three, and is shown in Figure 11.

---

**user:** Output a series of actions to transform the knot from its initial configuration to the goal gauss code.
**user:** <image>
**user:** Goal specification: The conversion is considered successful, when the current knot has the same gauss code as the goal knot. When determining the gauss code, always start from the white segment and traverse the rope towards the red segment, record positive for over-cross and negative for under-cross. A visual example is included in the image. An flat loop has gauss code of [].
**user:** Action specification: We follow a right-hand coordinate system, centered in the figure. Each action is in the form of [x,y,z,fx,fy,fz] where (x,y,z) are 3D coordinates which will be rounded to the closest rope segment, and (fx,fy,fz) are force vectors to be applied to that rope segment. x,y,z,fx,fy,fz are floating points bounded by [-1, 1]. In the image are three examples of before-and-after pairs of the unit directions. Use them as a reference. You can compose an action, for example, [1.0,1.0,0.0,0.9,0.0,-0.7] means pulling the most upper-right segment with 0.9 unit force in +x direction and 0.7 unit force in -z direction. You can select a segment somewhere in the middle of the rope, for example, let [x,y,z]=[-0.5,0.5,0.0] would be in the center of second quadrant.
**user:** Now consider a goal knot of the goal gauss code (what's the gauss code of the following knot?):
**user:** <image>
**user:** Here is the current knot:
**user:** <image>
**user:** What are a series of actions that will transform the current knot such that it has the same gauss code as the goal knot? Think step by step, and end your answer in one <answer></answer> block, like: <answer> [-0.8, 0.8, 0.0, 0.9, 0.0, 0.0] [0.0, 0.2, 0.0, -0.7, 0.0, 0.0] </answer>. You can include multiple lists of six floats in the block, separated by new lines.

---

Figure 8: Example open prompt and response.

---

**user:** Output a series of actions to transform the knot from its initial configuration to the goal gauss code.
**user:** <image>
**user:** Goal specification: The conversion is considered successful, when the current knot has the same gauss code as the goal knot. When determining the gauss code, always start from the white segment and traverse the rope towards the red segment, record positive for over-cross and negative for under-cross. A visual example is included in the image. An flat loop has gauss code of [].
**user:** Action specification: We follow a right-hand coordinate system, centered in the figure. Each action is in the form of [x,y,z,fx,fy,fz] where (x,y,z) are 3D coordinates which will be rounded to the closest rope segment, and (fx,fy,fz) are force vectors to be applied to that rope segment. x,y,z,fx,fy,fz are floating points bounded by [-1, 1]. In the image are three examples of before-and-after pairs of the unit directions. Use them as a reference. You can compose an action, for example, [1.0,1.0,0.0,0.9,0.0,-0.7] means pulling the most upper-right segment with 0.9 unit force in +x direction and 0.7 unit force in -z direction. You can select a segment somewhere in the middle of the rope, for example, let [x,y,z]=[-0.5,0.5,0.0] would be in the center of second quadrant.
**user:** Now consider a goal knot of the goal gauss code (what's the gauss code of the following knot?):
**user:** <image>
**user:** Here is the current knot:
**user:** <image>
**user:** What is the next action to take? Think step by step, and end your answer in one <answer></answer> block, like: <answer> [0.0, 0.2, 0.0, -0.7, 0.0, 0.0] </answer>. You should only include one list of six floats in the block.

---

Figure 9: Example stateless prompt and response.

Figure 10: Example stateful prompt and response.

# B   Benchmarking Details

This sections details to help with reproducibility. When a hyperparameter is not specified, we use the default value from the official implementation. Deviations from the default settings are **bolded**.

## B.1   PPO

We use the Stable-Baselines3 [Raffin et al., 2021] implementation of vision PPO. Table 6 lists hyperparameters values. The total number of trainable model parameters is 16.5 M. We have 32 vectorized environments. Each training run was conducted on 32 CPU cores, 96 GB RAM, and one NVIDIA GTX 2080 Ti GPU (11GB). Each training run took around 4 hours.

## B.2   DreamerV3

We use the official code base released by Hafner et al. [2025a]. We use the version with 12M trainable parameters, roughly the same size as the PPO model-free policy. We use the exact same default hyperparameters as the released repository (Table 7). We have 32 vectorized environments. Each training run was conducted on 33 CPU cores, 125 GB RAM, and one NVIDIA H100 GPU (80GB). Each training run took around 14 hours.

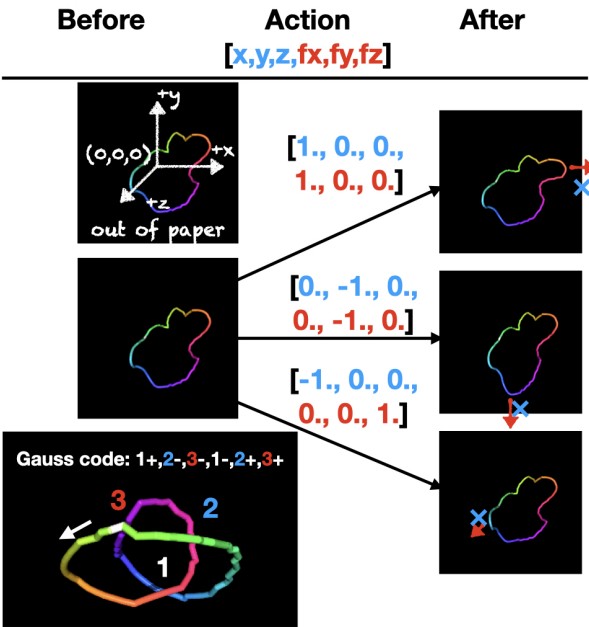

Figure 11: The first image of the user prompt. This image is used to introduce the task structure (together with text instructions), and to prime the model of system dynamics.

Table 6: PPO hyperparameters.

| Name | Value | Comment |
|------|-------|---------|
| Learning rate | **1e-5** | Sweeping over {1e-5, 3e-5, 1e-4, 3e-4 (default)}. |
| Feature dimension | **768** | Sweeping over {512 (default), 768, 1024} |
| Batch size | 64 | Default |
| Number of epochs | 10 | Default |
| Gamma | 0.99 | Default |
| GAE lambda | 0.95 | Default |
| Clip range | 0.2 | Default |

## B.3 TD-MPC2

We use the official code base released by Hansen et al. [2024]. We take the default architecture with 5M trainable parameters. Although we experimented with policies as large as the 48M model, we did not observe significant improvements over the default 5M model on KNOTGYM. Table 8 lists hyperparameters.

We alter the convolutional encoder because the original code base only supports $[c, 64, 64]$ observations, while we need $[3, 128, 256]$ for a fair comparison across methods. We added one additional pair of `nn.Conv2d(num_channels, num_channels, 5, stride=2)`, `nn.ReLU(inplace=False)` after the existing `Conv2d` layer of kernel size 5, and one fully connected linear layer between the last flatten layer and the final activation layer.

With considerations of training wall time, we use the experimental `vectorized_env` branch that supports parallel environments. We use 16 parallel environments. Each training run was conducted on 16 CPU cores, 148 GB RAM, and one NVIDIA GTX 2080 Ti GPU (11GB). Each training run took up to 20 hours.

Table 7: DreamerV3 hyperparameters.

| Name | Value | Comment |
| --- | --- | --- |
| Learning rate | 4e-5 | Default |
| Train ratio | 256 | Default |
| Hidden layer size | 256 | Default |
| Number of classes | 16 | Default |
| Replay buffer size | 5e6 | Default |

Table 8: TD-MPC2 hyperparameters.

| Name | Value | Comment |
| --- | --- | --- |
| Number of environments | **16** | Sweeping over { 1 (default), 2, 4, 8, 16, 32, 64} |
| Steps per update | **4** | Sweeping over { 1 (default), 2, 4, 8, 16} |
| $\rho$ | **0.7** | { 0.5 (default), 0.7 (suggested for episodic tasks)} |
| Learning rate | 3e-4 | Default |
| Batch size | 256 | Default |
| MPC | True | Default |
| Iterations | 6 | Default |
| Number of samples | 512 | Default |
| Number of elites | 64 | Default |
| Number of trajectories | 24 | Default |
| Horizon | 3 | Default |

## C KNOTGYM Implementation Details

**The Environment**  KNOTGYM uses the MuJoCo physics simulator [Todorov et al., 2012]. Each knot is modelled as a chain of beads (rigid bodies). The knot "floats" in a viscous medium – a performance related design decision to reduce collision checking between the rope and the plane. The pixel observation is generated by MuJoCo's default renderer. The camera is set to track the center of mass of the knot from a fixed distance and orientation. We sample all knot configurations with a combination of periodic saving on a random policy and manual filtering for configurations that are too twisted thus not suitable to be goal exemplars. We normalize the action space by default. Table 9 lists the environment specifications.

Table 9: KNOTGYM environment specifications.

| Name | Value |
| --- | --- |
| Max n crossings | One of {1,2,3,4} |
| Observation Space | Shape: [3, 128, 256], dtype: uint8 |
| Action Space | Shape: [6], dtype: float 32, range: [-1,1] |
| Frame skip | 24 |
| Reward for Gauss code equality | +5 |
| Punishment for timeout | -5 |
| Max episodic steps | 50 |
| Reset noise scale | 0.015 |

**Software Credit**  We use the following software: Stable-baselines3 [Raffin et al., 2021], DreamerV3 [Hafner et al., 2025a], TD-MPC2 [Hansen et al., 2024], PyKnotId [Taylor and other SPOCK contributors, 2017], Gymnasium [Towers et al., 2024], and MuJoCo [Todorov et al., 2012].

Stable-baselines3 (`https://github.com/DLR-RM/stable-baselines3`) has MIT license.

DreamerV3 (`https://github.com/danijar/dreamerv3`) has MIT license.

TD-MPC2 (`https://github.com/nicklashansen/tdmpc2`) has MIT license.

PyKnotId (`https://github.com/SPOCKnots/pyknotid`) has MIT license.

Gymnasium (`https://github.com/Farama-Foundation/Gymnasium`) has MIT license.

MuJoCo (`https://github.com/google-deepmind/mujoco`) has Apache license 2.0.

## D Additional Results

**Model Selection** We select the model closest to 1M environment steps for generalization analysis. How does training affect generalization? Figure 12 shows the generalization dynamics of a single `tie #X=2` training run with 20 goal configurations. While the train success rate increases drastically, the success rate evaluated on the test split configurations with same #X increases only modestly. This suggests that the policy has difficulty generalizing to different goals even with the same level of complexity (same #X).

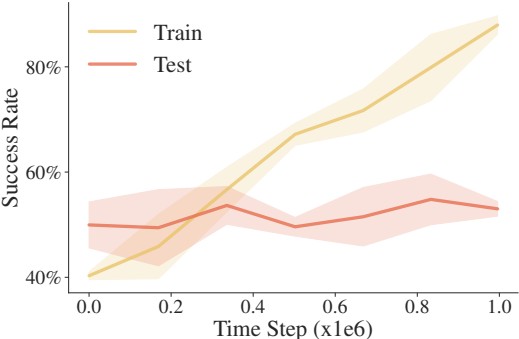

Figure 12: Generalization dynamics of a DreamerV3 `tie #X=2` training run. Error shades are 95% confidence intervals across three seeds.

**Training Curves** Figure 13 shows all RL training curves in the first million steps. For the `unknot` task, all methods manage to learn, despite different sample efficiency. For the harder `tie` and `convert`, none of the methods surpass random baseline when #X>2, showcasing the learning challenges KNOTGYM presents to RL methods.

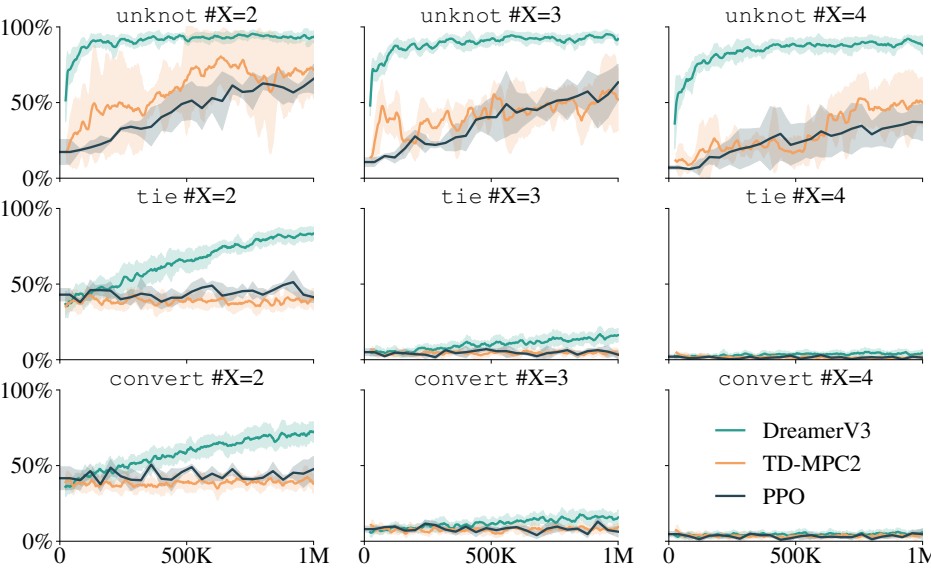

Figure 13: RL training curves: success rates on the train split versus number of steps across nine KNOTGYM tasks. Error shades are 95% CI computed over three seeds.

