# OpenReview forum: "Knot So Simple: A Minimalistic Environment for Spatial Reasoning"
_NeurIPS.cc/2025/Datasets_and_Benchmarks_Track — NeurIPS 2025 Datasets and Benchmarks Track poster_

### Official Review · Reviewer_8CAs · 2025-06-28

**Rating:** 5
**Confidence:** 2

**Summary:**

KNOTGYM is a new interactive Gym-compatible environment for pure-vision knot manipulation, designed to benchmark agents on three goal-conditioned tasks: unknot (reduce to a simple loop), tie (create a target knot), and convert (transform one knot into another), which are parameterized by a topological complexity measure. KNOTGYM supports controlled train/test and complexity-generalization ladders, and abstracts away end-effector details by directly simulating forces in MuJoCo. The authors benchmark model-free (PPO), model-based (DreamerV3), MPC (TD-MPC2), and chain-of-thought–prompted VLMs (GPT-4.1-nano), revealing that while RL methods solve “unknot” reliably (≈90% success at #X≤4), they struggle on “tie” and “convert” beyond #X=2, and VLMs largely underperform random baselines despite correct high-level reasoning.

**Dataset Code Accessibility:**

Yes

**Ethical Considerations:**

No, there are no or only very minor ethics concerns

**Final Justification:**

1. My previous concern about complexity ceiling is evidence-based. In the authors' statement, all state-of-the-art RL agents already fail when #X=4. So the author start where learning is still measurable, yet KnotGym can be extended to any #X. (e.g., #19, #20, ...).

2. As for the human study, I agree with the author and reviewer that it is hard to be properly collected. Since the author has justified their benchmark at this point and they plan to extend this in the next research, I think it is acceptable at this point.

In summary, the authors have justified their benchmark scope by setting an obtainable goal for current RL methods, and their environment design decisions is minimal, general, and most importantly, extensible. I think this paper is solid and can be accepted.

**Limitations Weaknesses:**

1. Limited Complexity: The authors stop at #X=4 crossings. I’d love to see trefoils or figure-eights—real-world knots are messier.

2. Flat Images: It’s purely 2D RGB. Adding depth or multi-view would match real rope-perception challenges and avoid ambiguity.

3. No Human Study: It’d be great to get people’s take, do the learned knot sequences look natural or efficient to us?

**Strengths Contributions:**

1. Clear Difficulty Scale: Using the crossing number as a “difficulty knob” is neat—it makes it straightforward to see how well agents scale up.

2. Plug-and-Play API: It’s just a standard Gym environment, so you can drop in your favorite RL or planning algorithm without wrestling with custom interfaces.

3. Pixel-Precision: Knots can cross over just a few pixels, so this really pushes an agent’s visual acuity and planning chops more than, say, Ant-Maze.

---

> ### Author Rebuttal · Authors · 2025-07-30
>
> We thank the reviewer for their positive feedback and helpful suggestions. In response to the limitations and weaknesses, we will clarify our benchmark scope as well as some environment design decisions.
>
> > Limited Complexity: The authors stop at #X=4 crossings. I’d love to see trefoils or figure-eights—real-world knots are messier.
>
> We capped preliminary benchmarking experiments at #X=4 because it is already too hard for existing RL methods: we only observe <5%, about-random policy learning at #X=4 for `tie` and `convert` across *all* methods.
>
> Trefoil belongs to #X=3 and figure-eight belongs to #X=4. On one hand, we cover reducible #X=3 and #X=4 knots in our experiments. But, we exclude trefoil-equivalent and figure-eight-equivalent knots because they cannot be reduced to fewer crossings as prime knots, so the intermediate knot has 3+ crossings throughout any episode, already proven hard on our selected RL methods.
>
> Therefore, we can safely expect at-best-random policy learning from current RL methods on #X>4 or non-reducible knots.
>
> While our experimental scope is informed by existing methods and aims to set an obtainable goal for laboratory AIs in the near future, the KnotGym simulator already supports up to #X=8, and the task specifications remain valid for any #X, 10 or 50 or 100. In context, the current frontier established by computational topology is at #X=19, which took months on a cluster.
>
> Burton, Benjamin A. "The next 350 million knots." In 36th International Symposium on Computational Geometry (SoCG 2020), pp. 25-1. Schloss Dagstuhl–Leibniz-Zentrum für Informatik, 2020.
>
> > Flat Images: It’s purely 2D RGB. Adding depth or multi-view would match real rope-perception challenges and avoid ambiguity.
>
> Regarding a depth channel: we focus on  2D RGB images, mostly because **how humans can manipulate a knot solely via a single RGB stream**. We also carefully tune 2D RGB images (e.g., image size, cable radius, coloring, lighting) in KnotGym such that there is sufficient information to reason about the state (e.g., the under/over crossings) without an additional depth channel. VLM-based methods are pretrained on 2D RGB inputs (no depth), so it would be hard to compare head-to-head in that sense.
>
> Regarding multiple camera views: Our ambition is towards *general* complex visual reasoning of which KnotGym is only one instance, so we are mindful of adding any domain-targeting feature such as multi-view, which we expect to help address domain-specific ambiguity.
>
> Nevertheless, KnotGym is a simulation environment **easily extensible** (as in less than 20 lines of code or mjcf changes) to an RGBD observation or multi-view cameras - should those scenarios be of interest to researchers.
>
> > No Human Study: It’d be great to get people’s take, do the learned knot sequences look natural or efficient to us?
>
> Excellent suggestion. Besides the evaluation aspect, collecting human manipulation trajectories also enables data-dependent solutions like behavior cloning or learning from demonstrations.
>
> While we are unable to properly collect data from multiple human subjects during this rebuttal period, we provide our (the author’s) subjective take on “do the learned knot sequences look natural or efficient to us?” Figure 6 showcases selected trajectories. In our opinion, the converged strategy for unknot (“pulling in one direction”) appears natural, surprisingly effective, but quite inefficient. In contrast, reviewer HGif considers it trivial and not a “genuine goal achieving” behavior. The sequences of the other two tasks (`tie`/`convert`) appear less interpretable or consistent. As humans, we (one of the authors) can solve any of the #X<5 tasks under 30 seconds. We will extend this type of survey for the next version of the paper.
>
>
> Hopefully, we have justified our benchmark scope (to set an obtainable goal for current RL methods in the near future) and our environment design decisions (to keep it minimal, general, and extensible). We agree with the aspect of human evaluation and will incorporate a more thorough version in the future version of this paper. Please kindly consider improving the overall score if our rebuttal helps.

---

> > ### Author Response · Authors · 2025-08-06
> >
> > Hi 👋
> > We completely understand these are busy times, but if you can give our response a look, we will greatly appreciate it. We tried really hard to answer all the concerns raised. Thanks! We are happy to discuss.

---

> > ### Comment · Reviewer_8CAs · 2025-08-06
> >
> > The authors addressed all my concerns. I decide to raise my score.

---

### Official Review · Reviewer_4u4o · 2025-06-29

**Rating:** 5
**Confidence:** 3

**Summary:**

This work proposes a new environment named KnotGym, including various goal-oriented rope manipulation tasks. These tasks are classical partial observation settings for complex, spatial reasoning and manipulation, of which the complexity is based on the number of knot crossings. To evaluate the complexity of this environment, this work evaluates various baselines, including model-free, model-based, and chain-of-thought reasoning methods.

**Dataset Code Accessibility:**

Yes

**Dataset Code Comments:**

This work has provided the code in an anonymous link, including environments and baselines.

**Ethical Considerations:**

No, there are no or only very minor ethics concerns

**Final Justification:**

Thanks for the author's response. I think this paper can raise more concerns in the field and is worth being accepted.

**Limitations Weaknesses:**

- Baselines of model-free methods are limited; it's better to include more baselines for POMDP, like DrQ, CURL, and so on.

- The rope sleeve seems to contain rich spatial information. Will the input of images lose a lot of information? What about the performance of baselines between the observation input and the state input?

--------

**Thanks for the author's response. I think this paper can raise more concerns in the field and is worth being accepted.**

**Strengths Contributions:**

- I really like the idea of designing the knot environments, which is easy to generalize, contains rich spatial/topological information, requires reasoning, and has potential for future real-world robot manipulation to untie the knot.

- Including LLM-based baselines is interesting and important for future research about the spatial reasoning ability of LLM.

---

> ### Author Rebuttal · Authors · 2025-07-31
>
> We thank the reviewers for their positive feedback. In response to the limitations and weaknesses:
>
> > Baselines of model-free methods are limited; it's better to include more baselines for POMDP, like DrQ, CURL, and so on.
>
> We attach preliminary training episode success rates of DrQ-v2 (model-free) with comparison with DreamerV3 (model-based), in the same format as Figure 5.
>
> | Task          | DrQ-v2 @200k steps | DreamerV3 @200k steps |
> |----------------|------------------|------------------------|
> | unknot #X=2    | 39.6             | 90.4                   |
> | unknot #X=3    | 38.5             | 89.1                   |
> | unknot #X=4    | 32.1             | 80.6                   |
>
> Given the same frame budget, it seems DrQ-v2 is much less sample efficient than DreamerV3, echoing conclusions from DrQ-v2 paper on sparse binary reward tasks (Figure Turn Hard). We are unable to conclude convergence yet: We snapshot at 200k frames because DrQ-v2 does not support parallel environments by default. This led to slower experimentation because KnotGym has a lower frame rate than typical DMControl tasks due to the high DoF of ropes. (We are actively improving rope simulation speed as well.) Additional experiments are ongoing, and we will report them when available or in the final version of the paper.
>
> > The rope sleeve seems to contain rich spatial information. Will the input of images lose a lot of information? What about the performance of baselines between the observation input and the state input?
>
> *We are not completely sure about the rich information referred to by this question, so if our answer misses the mark, we will appreciate further clarification of the reviewer’s intent here.*
>
> Compared to direct state inputs (i.e., coordinates of each bead), rendered rope sleeves introduce some occlusion, but can be mostly mitigated by tuning cable radius and increasing pixel resolutions. We chose 128x128 (an increase from 84x84 in default DMControl) by eyeballing whether the under/over crossings are legible. Our choice balances throughput vs. details. Higher resolution will be necessary in more complex settings (large #X)
>
> We use RGB inputs because we wanted to align as much as possible with the predominance of visual input in human processing of spatial reasoning tasks. The ground-truth coordinates are often elusive in real-world planning: a rope is unlikely to arrive with odometry sensors, and we would most likely rely on state estimation from RGB/D cameras. Also, VLM-based methods are pretrained on RGB inputs, not 3D coordinates, so it would be hard to compare head-to-head in that sense.
>
> Nevertheless, KnotGym is a simulation environment, so the underlying state is always accessible. Our code allows for the state input via the argument “output_pixels=False”, should those scenarios be of interest to researchers.
>
> Below are preliminary results for state-based PPO. The new observation space is (2x3x100+3,) by concatenating current knot coordinates, goal knot coordinates, and the action taken. We made necessary changes that are not thoroughly ablated/swept due to time constraints:
>
> - Increase learning rate from 1e-5 to 1e-4.
> - Replace CNN feature extractor with MLP feature extractor.
> - Normalize the state and force the center of mass at the origin.
>
> The table below is formatted similarly to Table 5 in the PDF, with each entry representing the training episodic success rate (higher the better).
>
> | Task           | PPO vision | PPO state |
> |----------------|------------|------------|
> | unknot #X=2    | 65.7       | 95.9       |
> | unknot #X=3    | 63.3       | 84.1       |
> | tie #X=2       | 41.3       | 43.5       |
> | tie #X=3       | 3.3        | 1.3        |
> | convert #X=2   | 47.7       | 50.1       |
> | convert #X=3   | 6.3        | 7.6        |
>
> For the `unknot` task, data efficiency of state-input solutions stands out, which agrees with the literature. This edge disappears for harder, goal-conditioned `tie`/`convert` tasks: PPO vision and state are practically the same within error margins. We think this is because learning to infer abstract topological goals from coordinates is no simpler than from pixel inputs. This is intriguing when compared to how pretrained VLMs seem to possess such capability zero-shot.
>
> Hopefully we have addressed the concerns about other model-free baselines and state-based inputs. Any follow-ups are welcome.

---

> > ### Comment · Reviewer_4u4o · 2025-08-06
> >
> > Thanks for your response, which has fully addressed my concerns. I think this paper can raise more concerns in the field and is worth being accepted.

---

### Official Review · Reviewer_HGif · 2025-07-03

**Rating:** 5
**Confidence:** 4

**Summary:**

This paper introduces **KNOTGYM**, an interactive environment designed to study complex spatial reasoning and rope manipulation tasks. The environment defines goal-directed manipulation problems using knot representations characterized by varying numbers of crossings, forming a natural axis of complexity and a well-structured generalization ladder. KNOTGYM provides a platform to evaluate agents’ abilities in spatial reasoning, action planning, and abstraction. The authors benchmark a range of representative methods and highlight the challenges presented by the environment.

**Dataset Code Accessibility:**

Yes

**Ethical Considerations:**

No, there are no or only very minor ethics concerns

**Final Justification:**

Thank the authors for the response. I have read all the response, and I think my confusions have been addressed.

**Limitations Weaknesses:**

1. **Lack of Failure Case and Limitation Analysis**.

   The paper does not distinguish whether the model failures are due to perceptual ambiguity or reasoning challenges, which limits the insight into what the benchmark is truly testing.

2. **Limited Real-World Applicability**.

   While the benchmark is novel and interesting, it remains somewhat distant from real-world applications. The simulation environment is highly idealized, lacking the sensory noise, physical constraints, and variability typically present in practical scenarios. As a result, it is unclear whether models trained and evaluated in this setting would generalize to physical systems or be applicable in realistic tasks. Bridging this gap—through more realistic simulation settings or demonstration of sim-to-real transfer—would enhance the benchmark’s practical relevance.

3. **Insufficient Robustness Evaluation**
   The current benchmark design does not sufficiently encourage robust policy learning. Specifically, as described in Line 139, the episode terminates upon a single match with the target Gauss code, which can lead to trivial solutions that exploit momentary alignments rather than achieving stable, meaningful manipulation strategies.
   This concern is clearly illustrated in Figure 6, where agents converge to a simple "pull in one direction" behavior. Such behaviors may exploit the termination criterion to appear successful without demonstrating genuine goal achievement or stability.

   **Suggestion**:

   - Revise the benchmark to require that the goal configuration (i.e., Gauss code match) be maintained for a consecutive number of steps (e.g., 5–10 timesteps) before termination, or alternatively, encourage agents to **maximize the duration** for which the goal configuration is maintained.
   - Additionally, report the **average episode length upon success** to measure policy robustness and temporal consistency.
   - Reproduce results in Figure 4 and Figure 6 using the revised termination criterion to verify the impact on learned behaviors.

**Strengths Contributions:**

- **Novel and Well-Motivated**: KNOTGYM is a novel and well-justified environment targeting spatial reasoning—a fundamental yet underexplored challenge in embodied intelligence. The setup is clean, interpretable, and pushes the boundary of what current learning-based methods can handle.
- **Theoretical Clarity**: The use of Gauss codes to define knot equivalence classes provides a principled mathematical foundation for both task formulation and evaluation, enabling systematic generalization analysis.
- **Clarity and Presentation**: The paper is well-written, logically structured, and easy to follow. The figures and descriptions effectively convey the environment's design and challenges.
- **Simplicity and Elegance**: Despite the complexity of the task domain, the proposed framework remains conceptually simple and elegant, offering a useful testbed for broader research in spatial and topological reasoning.

---

> ### Author Rebuttal · Authors · 2025-07-30
>
> We thank the reviewer for both their positive feedback and constructive advice. In response to their review:
>
> > Lack of Failure Case and Limitation Analysis.
>
> We provide quantitative results (Table 6) in Section 4.1 and 4.2, where we offer insights on failure cases uniquely diagnosed by KnotGym. For example, goal-conditioned tasks like `tie` and `convert` challenge all RL methods (Line 220), indicating that complex goal generalization is a challenging task. We further hypothesize and verify that insufficient in-distribution exploration limits best-performing RL training (Line 234).
>
> > The paper does not distinguish [...] [between] perceptual ambiguity or reasoning challenges
>
> KnotGym is similar to existing benchmarks in its design. For example, Gym’s MuJoCo environments (Ant, Half Cheetah, Humanoid, etc) – while it’s hard to disentangle failure causes in these environments, they have nevertheless been extremely influential and have been the driving force behind RL algorithmic development in recent years (e.g., PPO). KnotGym exposes new challenges beyond these benchmarks.
>
> We acknowledge the importance of compartmentalized error analysis. KnotGym is an environment compatible with all kinds of solutions, from end-to-end RL (more opaque) to using vision language models, or VLMs (more interpretable). To maximize the scope of applicable solutions, we make minimal assumptions in the solution space (model-free? neural-symbolic components? explicit world models?). As a consequence, we conduct error analysis of each benchmarked method in a case-by-case manner, when the solution is more interpretable, e.g., VLMs can perceive and reason well, but fail to conduct grounded actions (Line 264, and Figure 7).
>
> > …which limits the insight into what the benchmark is truly testing
>
> The KnotGym benchmark tests all of the above, with particular emphasis on the integration of acute perception, complex planning, and grounded action. We think these three modular steps are intertwined and should inform each other in the context of long-horizon reasoning: How does the planning module account for all sources of uncertainty, let it be perceptual/sensory, epistemic, or from flawed execution? Can a solution perform chain-of-thought style spatial reasoning despite imperfect world models and cascading errors? These remain open research questions, and KnotGym offers a lightweight yet challenging testbed that enables early iterations towards answering them.
>
> > Limited Real-World Applicability. [...] unclear whether models trained and evaluated in this setting would generalize to physical systems or be applicable in realistic tasks [...]
>
> KnotGym is a diagnostic task focused on fundamental challenges. In this sense, it’s similar to the Gym MoJuCo environments we point to above. In a similar fashion, the target is not real-world fidelity. Past experience with MoJuCo environments shows how impactful such environments are, even when they are not ultra-high-fidelity. Our research motivation is not to find a set of model weights that would tie a perfect knot in a real-world context, but rather to **create a testbed to study fundamental challenges (complex visual reasoning), and do it in a research accessible way** – hence the use of fast simulation. The knot domain enters the scene because knot theory offers a sound notion of complexity, making it a good instantiation of more complex visual reasoning tasks that we anticipate future AIs to solve.
>
> We think the analogy to existing benchmarks with proven track records is particularly useful. Please consider cartpole, MuJoCo Ant/Cheetah/Humanoid/etc, and their immense impact. Key to this was research accessibility. We keep the underlying knot simulator lightweight and minimal as long as it implements the knot manipulation task and passes the eye-ball test. More performance tradeoffs are made in a similar spirit, e.g., we drop gravity and a support surface like a table to reduce unnecessary collision checking. There is other work (such as Yan et al. 2020) that focuses on sim2real transfer, which is an equally important topic, though tangential to our contribution - a lightweight diagnostic testbed exhibiting factorial complexity scaling.
>
> Yan, Mengyuan, Gen Li, Yilin Zhu, and Jeannette Bohg. "Learning topological motion primitives for knot planning." In 2020 IEEE/RSJ International Conference on Intelligent Robots and Systems (IROS), pp. 9457-9464. IEEE, 2020.
>
>
> > Insufficient Robustness Evaluation. [...] revising termination criterion [...] lead to trivial solutions that exploit momentary alignments rather than achieving stable, meaningful manipulation strategies [...]
>
> Unlike maintaining cartpole balance, KnotGym tasks have a natural notion of a final step, like navigating a maze. Our specification is consistent with existing RL environment implementations, for example, Farama Foundation’s AntMaze (sparse) – an episode terminates immediately with a single reward of 1.
>
> We intentionally tune down the control frequency in KnotGym simulator (control timestep = 0.008 * 24 = 0.192 sec). So actions in KnotGym are coarse compared to typical continuous control tasks (DMControl Humanoid has a control timestep of 0.025; AntMaze is 0.010.) This design is intended to prevent momentary alignment due to high-frequency control signals.
>
> **That said, we do offer an option (via “done_after=5” argument) to set the episode termination** upon five consecutive steps of holding the goal Gauss code. Setting done_after=inf` transforms it to a continuing task, if one desires so. Each frame that satisfies the goal Gauss code yields +1 reward. This complicates analysis because an inefficient policy may acquire more return than an efficient one (010110111011111[term] versus 011111[term]). We report preliminary PPO average episode success rates in the same condition as Table 5 below. "Orig." indicates the original done_after=1 configuration; "Alt." indicates the alternative done_after=5 configuration. Due to rebuttal time constraints, Alt. columns were run on 1 seed only as opposed to averaging across 3 seeds for the Orig. columns.
>
> | Task           | Orig. random | Orig. PPO                      | Alt. random | Alt. PPO                      |
> |----------------|--------------------------|-----------------------|--------------------------|-----------------------|
> | unknot #X=2     | 11.1                     | 65.7                  | 7.1                      | 53.8                  |
> | unknot #X=3     | 8.4                      | 63.3                  | 5.1                      | 34.6                  |
> | tie #X=2        | 35.9                     | 41.3                  | 13.1                     | 14.1                  |
> | tie #X=3        | 2.5                      | 3.3                   | 0.9                      | 1.0                   |
> | convert #X=2    | 36.8                     | 47.7                  | 11.6                     | 14.7                  |
> | convert #X=3    | 9.2                      | 6.3                   | 3.5                      | 1.8                   |
>
> The random/PPO train success rates in the alternative column are consistently lower than their corresponding values in the original column, suggesting that the “hold” action presents a non-trivial learning challenge under the original training protocol (specifically, more steps might help, but we kept 1M steps for consistency). Qualitatively, we observe that setting done_after=5 tends to converge to the same “pulling in one direction” unknotting strategy (in the easiest of the KnotGym challenges).
>
> > [...] This concern is clearly illustrated in Figure 6, where agents converge to a simple "pull in one direction" behavior. Such behaviors may exploit the termination criterion to appear successful without demonstrating genuine goal achievement or stability.
>
> We observe the  “simple pull in one direction” behavior in Figure 6, even when PPO uses done_after=5 (similar to the reviewer suggestion). There are two reasons for the convergence:
>
> - Such a “trivial” solution exists for small #X knots. You can verify this in real-life by gently shaking or tossing lightly tangled shoelaces or a pair of headphones.
> - The policy does not need to reason about the goal for each episode, so experiences across episodes generalize better (Line 222). In contrast, the other two tasks `tie` or `convert` are goal-conditioned, therefore efficient exploration is more of a challenge.
>
> As such, we consider the `unknot` task as **a sanity check** (we will emphasize this in the paper), and the goal-conditioned `tie`/`convert` tasks are significantly more challenging.
>
> In general, we focus on learning a policy that *can* reach the goal, whether by exploration or exploitation, which already proves to be hard enough for `tie` and `convert` when #X>2. It is perfectly valid and valuable to ask how to guide the policy to learn a more efficient/sophisticated/human-like solution. As we open-source a minimalistic base environment, researchers can easily add wrappers or adapt the environment as they wish towards their definition of “genuine” goal achievement. **The openness of KnotGym is a key feature, and we are looking forward how researchers customize it.**
>
> > Additionally, report the average episode length upon success to measure policy robustness and temporal consistency.
>
> We agree that average episode length upon success is an informative metric, as it measures solution efficiency. We will summarize them in a tabular form in the final version.
>
> Hopefully we have clarified misunderstandings about our design motivation, which seems to be the root of other concerns. Please kindly consider improving the overall score if our rebuttal helps.

---

> > ### Author Response · Authors · 2025-08-06
> >
> > Hi 👋
> > We completely understand these are busy times, but if you can give our response a look, we will greatly appreciate it. We tried really hard to answer all the concerns raised. Thanks! We are happy to discuss.

---

### Decision · Program_Chairs · 2025-09-18

**Decision:**

Accept (poster)

**Comment:**

(a) The paper proposes a new environment/benchmark for spatial reasoning and rope manipulation.

(b) The benchmark is interesting, relevant, novel, well-motivated, easily extendable, and timely. The theoretical underpinnings are well done. The paper is well written. The benchmark is likely to be picked up by the community.

(c) There were some concerns regarding limitations regarding the evaluation metrics (failure cause, robustness), questions about real-world applicability, limited algorithms, various design choices (e.g. 2D images), and comparison to human performance.

(d) All reviewers vote for accepting the paper. All issues have been successfully addressed. The environment/benchmark itself is very well done and the paper provides interesting insights.

(e) All major concerns have been addressed successfully - some were misunderstandings and overlooked details, some have been addressed with additional details and results, some are promised to be included (seems very feasible), and for some others the authors managed to successfully explain and defend their design choices.